# Dynamics of care and sector use between birth, contraception and sick child services

Lindsay M. Mallick[1,2,3]*, Nicole Bellows[1], Rebecca Husband[4], Michelle Weinberger[1]

1 Avenir Health, Glastonbury, Connecticut, United States of America, 2 Department of Family Science, School of Public Health, University of Maryland, Maternal and Child Health Program, College Park, Maryland, United States of America, 3 Maryland Population Research Center, College Park, Maryland, United States of America, 4 Population Services International, Washington, District of Columbia, United States of America

* lmallick@terpmail.umd.edu

## Abstract

Governments in low- and middle-income countries increasingly recognize their role as stewards of mixed health systems comprising both public and private actors, but policy often lacks a nuanced understanding of how individuals switch between these two sectors for their healthcare needs, especially for family planning (FP) and maternal, newborn and child health (MNCH). In this cross-sectional study, we used data collected by The Demographic and Health Surveys Program between 2014–2021 from eight countries (Afghanistan, India, Indonesia, Kenya, Malawi, Nigeria, Pakistan, and Uganda) to describe service and sector use among women with a recent birth, a need for FP, and a child under five years old experiencing an illness (N=53,014). We applied multivariable logistic regressions in each country to test the associations first between sector for birth and missed opportunities for contraceptives or sick child care, and next, between sector use dynamics between birth and contraceptive use (sector fidelity, sector switching, or nonuse of services) and nonuse of sick child care. Sector at facility for birth and sector switching between services was common, but neither were generally associated with missed opportunities for care for sick children. However, private sector use at birth predicted nonuse of contraceptives in four countries, though the directionality varied. Consistently, women who did not access care at birth and contraception had significantly greater odds of missing care for sick children. Notably, in Malawi, the adjusted odds ratio of missed sick child care among those with nonuse of care for birth or contraceptives was 2.5 times that of those with sector fidelity (95% Confidence Interval: 1.43-4.22). These findings underscore the need for health system stewards to closely consider both the public and private sectors in health governance. Greater cross-sectoral cooperation and continuity of care is paramount to improving health outcomes.

**Data availability statement:** Data from this study were collected through household surveys administered by The DHS Program in Afghanistan (2015), India (2019-21), Indonesia (2017), Kenya (2014), Malawi (2015-16), Nigeria (2018), Pakistan (2017-2018), and Uganda (2016). Authors accessed these data from https://dhsprogram.com/, but at the time of article publication, the data were no longer accessible.

**Funding:** MOMENTUM Private Healthcare Delivery (MPHD) was funded by the U.S. Agency for International Development (USAID) as part of the MOMENTUM suite of awards and implemented by Population Services International (PSI) with partners Jhpiego, FHI360, ThinkWell and Avenir Health under USAID cooperative agreement #7200AA20CA00007. The contents of this report are the sole responsibility of PSI and do not necessarily reflect the views of USAID or the United States Government. Rather, MPHD abides by the USAID's Private Sector engagement principles to partner with governments and local systems to foster sustainable solutions to provide high-quality integrated healthcare. The research priorities of MPHD were developed through cross-country partnerships.

**Competing interests:** The authors have declared that no competing interests exist.

## Introduction

Increasingly, the global community recognizes that a life course approach to the family planning (FP), reproductive health (RH) and maternal, newborn, and child health (MNCH) needs of women in low- and middle-income countries (LMIC) is critical to achieving improved health outcomes [1,2]. These needs can be addressed by leveraging the complementary strengths of the public and private sectors [3]. To do so, a better understanding of healthcare seeking journeys across the public and private sectors is crucial for health policy and planning [4–6].

The public-private mix for FP and MNCH services varies substantially across countries. Several studies and resources detail these differences for individual health areas [6–11]. Within East Africa, private sector use among modern contraceptive users ranges from 7% in Rwanda to 40% in Uganda [11]. In West Africa, the use of the private sector for child health visits ranges from 19% in Senegal to 60% in Nigeria [11]. A study that examined the public-private mix for antenatal care (ANC), institutional birth, and postnatal care (PNC) in 22 large African cities found that patterns of care seeking in the public and private sector varied across cities, with no consistent pattern in use [12]. Within countries, the public-private mix also varies across individual health services. An analysis of Demographic and Health Survey (DHS) data from 70 LMICs from 1990-2013 indicated generally higher levels of private sector use for sick child visits for diarrhea or fever than for ANC, facility birth, or modern FP [10].

These variations in patterns of public and private sector service utilization across and within countries may be the result of multiple factors. Patterns of health service use across sectors vary by individual characteristics, for example, residence or wealth. Women living in urban areas and those with higher incomes use the private sector at higher rates than those living in rural areas and those with lower incomes [8,10,13,14]. Interpersonal and community factors, such as influence from husbands, family members, and neighbors can affect women's perceptions and decisions about care-seeking and choice of facility [15,16]. Other factors relating to these patterns include the types of facilities within the public and private sectors, country policies, the out-of-pocket costs of private sector services, and accessibility of private facilities [17,18]. While these previous studies highlight aggregate differences in public and private sector use for health services, they do not examine patterns of sector use across health service utilization for individuals. This type of analysis can reveal important information, as individuals may access some services in the private sector and others in the public sector. For example, a woman may use the public sector for ANC but switch to a private provider for birth, postpartum contraception, and subsequent child health visits.

Switching sectors could have implications for continuity of care over the life course, from contraceptive use during adolescence and other life stages, through pregnancy, childbirth, postpartum care, and health services for children [19]. The reasons for switching sectors are likely highly individualized and context-specific, though there is a dearth of research exploring this. For example, individuals may switch sectors because they prefer a certain provider, or because they have been referred to a provider in a different sector (and have the financial and transportation means to reach that provider). Conversely, sector switching could indicate

a discontinuity of care or fragmented care—due to negative experiences, inadequate resources, or lack of options for continuing care in the same sector. These experiences may hinder future healthcare seeking, leading to missed opportunities for follow-up and referrals, and potentially worsening health outcomes if care is delayed or not accessed. Though sector fidelity does not guarantee continuity of care, it may also facilitate future care seeking and result in fewer missed opportunities. However, the dynamics and implications of health seeking behavior across sectors are not well understood. An improved understanding of these patterns of care seeking can inform strategic communication interventions and guide FP and MNCH investments across health systems for maximum impact.

To address the gaps in the literature, we analyzed Demographic and Health Survey (DHS) data to examine sector use and sector fidelity at the individual level for childbirth, modern contraceptives, and sick child care across eight countries in Asia and sub-Saharan Africa.

The three research questions addressed by these analyses are as follows:

1. What are the patterns of sector use, sector switching, and missed opportunities for care across childbirth, modern contraceptives, and sick child care?

2. Is sector for birth associated with subsequent missed opportunities for using modern contraceptives and seeking sick child care?

3. Are sector fidelity and switching sectors across birth and contraception associated with subsequent missed opportunities for sick child care?

## Methods

### Data

Our analysis drew from publicly available DHS data. DHS surveys are designed to collect data from a nationally representative sample of households in LMICs using a multistage cluster sampling design. The surveys collect information on demographics, contraceptive use, and maternal, newborn and child health-related behaviors and outcomes, including healthcare seeking across multiple health areas.

We explored an initial set of 21 United States Agency for International Development Population and Reproductive Health priority countries that have conducted DHS surveys since 2014. Of these 21 countries, we selected the eight countries with the largest possible analytic sample to maximize the reliability and precision of the estimates produced by the analysis. This included four countries in sub-Saharan Africa and four in Asia. The eight countries and corresponding year of DHS data collection were Afghanistan (2015), India (2019–21), Indonesia (2017), Kenya (2014), Malawi (2015–16), Nigeria (2018), Pakistan (2017–2018), and Uganda (2016). The most recent survey available at the time of the analysis was used for each country.

### Sample

The analytic sample included ever-married women who were potentially in need of all three services: facility birth, modern contraception, and sick child services. Specifically, women were included if they reported (1) a delivery in the previous five years; (2) a potential need for FP (excluding pregnant, infecund, or menopausal women and those who expressed desire for having another child in the next two years); and (3) at least one child under five years of age with symptoms of fever, acute respiratory illness (ARI), or diarrhea in the last two weeks. We excluded women with missing information on any exposure, outcome, or covariate.

### Measures

We constructed variables to describe sector use and missed opportunities for care for each of the three health services: birth, contraceptive use, and sick child care. The DHS uses different recall timeframes for survey questions about these

three services, which allowed us to sequence them for our analysis, starting with childbirth in the last five years, then current modern contraceptives, then sick child care (Table 1). We considered the most recent birth as the first point of service needed in the cascade of the three services assessed. Over 99% of women in the sample in all countries but Uganda (98%) had a birth prior to one month before the survey. Contraceptive use and related services were assumed to be the second point in the cascade as we can only gauge current contraceptive use and not contraceptive use prior to birth. More than 94% of modern contraceptive users in our sample reported that they began using their most recent method longer than a month ago, so this service was also largely expected to have preceded sick child care for children who experienced symptoms of illness in the two weeks before the surveys.

### Sector

We assessed the sector of health facility use for each service point of birth, modern contraceptive use, and sick child care, as summarized in Table 2 and described as follows.

**Birth.** We assessed the location of the most recent birth and classified the sector of birth as public, private, or other/none. The other/none category included women who gave birth at home or at other non-facility locations.

**Modern contraceptive use.** Women in our sample were either current users of a modern contraceptive method or had potential unmet need for modern contraception. We constructed variables describing the source of the current contraceptive method. We classified the sector from which women received their current contraceptive method as public, private, pharmacy/shop, and other/none. We ascribed community health workers to the sector in which they were designated by each country. While some countries consider pharmacies and certain types of shops to be part of

**Table 1. Demographic and Health Surveys (DHS) timeframe references for birth, contraceptive, and sick child service needs.**

| Health service | Timeframe for measurement in DHS |
| --- | --- |
| Birth | Births in the last five years (analysis focused on *most recent* if more than one birth within the last five years) |
| Modern contraceptive use | Current modern contraceptive use or current unmet need for modern contraception, typically reflecting use beginning prior to the month of the survey. |
| Sick child care | Child under age five who had symptoms of an illness including fever, ARI, or diarrhea in the last two weeks. |

**Table 2. Summary of service use and sector classification for birth, contraceptives, and sick child care.**

| Health Service | Public | Private | Pharmacy or shop | Other or none (e.g., a missed opportunity) |
| --- | --- | --- | --- | --- |
| Birth | ✓ | ✓ | n/a | • Home birth<br>• Other non-facility birth location |
| Modern contraceptives | ✓ | ✓ | ✓ | • Other source (e.g., traditional practitioners, friends, relatives)<br>• Traditional method<br>• No method (has unmet need) |
| Sick child care | ✓ | ✓ | ✓ | • Informal outlets (e.g., traditional practitioners, friends, relatives)<br>• No care (has unmet need) |

the private sector, other countries do not consider these sites as formal health providers. In our analysis, because of the importance of pharmacies and drug shops in providing contraceptives [11,20], we classified use of pharmacies and drug shops as a separate source category. The other/none category included women who received contraception from informal sources (such as friends, relatives, or traditional practitioners), women who were not using a modern method, or women who were using a traditional method (abstinence, withdrawal, or other country-specific traditional or folk methods). Women in the other/none category were considered to have a *missed opportunity for contraception*, one of the two outcomes for the second research question.

**Sick child care.** Women in our sample reported at least one child under five years of age with symptoms of fever, ARI, or diarrhea in the last two weeks. We constructed variables for source of sick child care and missed opportunities for sick child care based on care seeking for children with these symptoms in the two weeks preceding the DHS survey. Source of sick child care was classified as public, private, pharmacy/shop, and other/none. Like contraceptive use, we classified use of pharmacies and drug shops as a category separate from the private sector and used country-specific designations to classify community health workers by sector. The other/none category included those who did not seek care and those who sought care at informal outlets like traditional practitioners, friends or relatives. If more than one child needed care or if care was obtained in more than one sector, the variable was coded as other/none if any single child did not receive care. Otherwise, the variable was coded hierarchically in the following order:

- *Private* if any care from a private facility;

- *Public* if no private facility care but any public facility care; followed by

- *Pharmacy/shop* if pharmacy/shop only and no public or private facility care.

Women in the other/none category were considered to have a *missed opportunity for sick child care*, an outcome for the second and third research questions.

## Sector fidelity between birth and contraception

We constructed a five-category variable for sector and use of health care patterns between the first two services in the cascade: childbirth and contraceptive use; this variable served as the main exposure variable for the third research question. The categories were:

1. *Sector fidelity:* women with consistent sector use between childbirth and contraception (both public or both private).

2. Sector *switching:* women with a public or private sector birth who used a different sector (or a pharmacy/shop) to access contraceptives.

3. *Facility birth, no contraceptive use:* women who gave birth in a facility but who were not using a modern contraceptive method from either a public, private, or pharmacy/shop source.

4. *No facility birth, contraceptive use:* women who did not give birth in a facility but who were using a modern contraceptive method and received it from either a public, private, or pharmacy/shop source.

5. *No facility birth, no contraceptive use*: women who did not give birth in a facility and who were not using a modern contraceptive method from either a public, private, or pharmacy/shop source.

## Covariates

For our statistical analyses, we examined potential confounders that have been shown to be associated with reproductive and maternal health care and sector use, including demographic and maternal characteristics, as well as indicators of access and availability of health care services [7,11,17,21]. These variables included place of residence (urban or rural),

wealth (tertiles of low, medium, or high, created from a continuous wealth score), education (none/primary vs. secondary and higher), parity (one child vs. two or more). Indicators of access to care included use of ANC during the most recent pregnancy that resulted in a live birth (0–3 visits vs. 4 or more visits), and any problem accessing health care related to securing money or permission for care, distance to care, or not wanting to go alone. We also examined cesarean section for the most recent live birth since cesarean sections are more common in the private sector than public [22]. In areas where the cesarean section rate is low, having had a cesarean section may indicate better access to services [23].

## Analysis

### Patterns of sector use, sector switching, and missed opportunities

To address the first research question, we produced Sankey diagrams, which are graphs that demonstrate the flow from one set of values to another across different stages [24]. The Sankey diagrams visualize the use of services and the sector within each service across the cascade, demonstrating the relative utilization of each sector and revealing missed opportunities for care.

### Sector for birth and subsequent missed opportunities

For the second research question, we conducted a descriptive analysis of differences in missed opportunities for contraception and sick child care according to the sector of delivery for the most recent birth. To explore the relationships between sector use for the most recent delivery and missed opportunities for contraceptive use or sick child care, we conducted multivariable logistic regression analyses. The outcome for one model was the dichotomous variable for missed opportunity for contraception, and for the other model, missed opportunity for sick child care. The main exposure variable for each model was the categorical variable describing the sector of birth for the most recent birth.

### Sector use patterns and subsequent missed opportunities for care

For the third research question, we conducted multivariable logistic regression analyses with the dichotomous variable for missed opportunity for sick child care as the outcome. The main exposure variable for this model was the categorical sector fidelity variable that describes sector use across delivery and contraception.

All models were adjusted for potential confounding variables, including residence, wealth, education, parity, ANC, cesarean section, and any problems accessing care. Estimates were weighted to adjust for nonresponse and disproportionate sampling. Statistical tests were adjusted for the complex survey design. All analyses were conducted using Stata version 16.1 [25].

## Ethical approval

The DHS Program receives approval through the ICF Institutional Review Board for all procedures and questionnaires. In each country where a DHS survey is implemented, a host country Institutional Review Board (IRB) reviews the survey protocol to ensure it complies with the norms and laws of the country. Interviewers read an informed consent statement to respondents who may accept or refuse to participate. For children or adolescents, a parent or guardian is required for the participation of children or adolescents. The DHS Program granted permission for use of its anonymized data for this study and further ethical approval for its use is not required by our institute.

## Results

### Overview of the sample

The analytic samples for each country ranged from 1,955 women in Kenya to 28,545 women in India. Among the sample of women in all countries with a recent birth, a potential need for contraceptives, and a recently sick child. More than half

**Table 3. Characteristics of the analytic sample of women with birth, contraceptive, and sick child service needs in the countries studied.**

| | Afghanistan 2015 | India 2019-21 | Indonesia 2017 | Kenya 2014 | Malawi 2015–16 | Nigeria 2018 | Pakistan 2017–18 | Uganda 2016 |
|---|---|---|---|---|---|---|---|---|
| | N(%) | N(%) | N(%) | N(%) | N(%) | N(%) | N(%) | N(%) |
| Rural residence | 3,787(70.8) | 20,326(73.7) | 2,792(53.4) | 1,234(63.1) | 4,182(86.5) | 1,554(58.7) | 1,387(62.1) | 2,601(81.9) |
| Poorest tertile | 1,717(32.1) | 8,931(32.4) | 1,455(27.9) | 433(22.2) | 1,737(35.9) | 749(28.3) | 576(25.8) | 999(31.5) |
| No/primary education | 4,814(90.0) | 8,619(31.2) | 1,525(29.2) | 1,309(66.9) | 3,900(80.7) | 1,485(56.0) | 1,280(57.4) | 2,362(74.4) |
| One child | 496(9.3) | 7,987(28.9) | 1,611(30.8) | 311(15.9) | 966(20.0) | 271(10.2) | 268(12.0) | 389(12.3) |
| Less than 4 ANC visits | 4,184(78.2) | 11,389(41.3) | 444(8.5) | 780(39.9) | 2,341(48.4) | 1,031(38.9) | 982(44.0) | 1,229(38.7) |
| Cesarean birth | 216(4.0) | 6,771(24.5) | 861(16.5) | 148(7.5) | 289(6.0) | 90(3.4) | 578(25.9) | 203(6.4) |
| Problems accessing care | 4,750(88.8) | 12,318(44.6) | 1,825(34.9) | 1,082(55.3) | 3,660(75.7) | 1,358(51.3) | 1,589(71.2) | 2,135(67.2) |
| Care for all three needs[1] | 867(16.2) | 8935(31.3) | 2822(54) | 661(33.8) | 2249(46.5) | 302(11.4) | 589(26.4) | 968(30.5) |
| Total (N) | 5,350 | 27,593 | 5,225 | 1,955 | 4,836 | 2,649 | 2,231 | 3,175 |

Note: ANC=Antenatal care; [1]Care for all three needs includes a facility for birth, modern contraceptive use, and care for a sick child.

of the sample in each country resided in rural areas. The percentage of women with no or primary education ranged from 29% in Indonesia to 90% in Afghanistan. Women in these two countries also demonstrated the lowest and highest percentages of less than four ANC visits (8.5% in Indonesia, 78.2% in Afghanistan) and lowest and highest report of problems accessing care (34.9% in Indonesia and 88.8% in Afghanistan). In all but one country, less than half of women received all three types of care, with a low of 11% in Nigeria and a high of 54% in Indonesia (Table 3).

## Patterns of sector use, sector switching, and missed opportunities

Using Sankey diagrams, Fig 1A illustrates sector use and missed opportunities for care across the sequence of health services in the four countries in Africa, and Fig 1B shows a Sankey diagram for the four countries in Asia. Use of the public sector predominated in three of the four sub-Saharan African countries—Kenya, Malawi, and Uganda—as well as in Afghanistan. The private sector had a relatively larger reach in three of the four countries in Asia (India, Indonesia, and Pakistan) for facility birth and sick child care. In Indonesia, the private sector was the source for the largest share of modern contraceptive users. Pharmacies played a significant role in contraceptive service provision in Afghanistan and Pakistan and in sick child care in Nigeria. Pharmacies were not frequently used in Kenya, Malawi, and Uganda. Nigeria, and to a lesser extent Afghanistan, were characterized by higher rates of non-use of health services and low rates of private sector service utilization.

Overall, switching sectors was common in all countries. In countries where non-use of health services was common, sector switching was particularly high among those who did seek care. This pattern was noted in Afghanistan and Nigeria. Sector fidelity occurred more frequently in countries where a single sector predominated services utilization. For example, sector fidelity was seen frequently in Malawi, Kenya, and Uganda where the public sector predominated. Sector fidelity was also higher in Indonesia, which has a relatively large private sector reach across services.

## Sector for birth and subsequent missed opportunities

Missed opportunities for contraceptive or sick child care were similar between those giving birth in the public and private sectors in most countries except Afghanistan and Nigeria (Fig 2). In Afghanistan, women who gave birth in the public sector appeared more likely to miss opportunities for contraception (58.7% vs. 38.9%); missed opportunities for sick child care were also slightly higher among women who gave birth in the public sector than in the private sector (39.0% versus

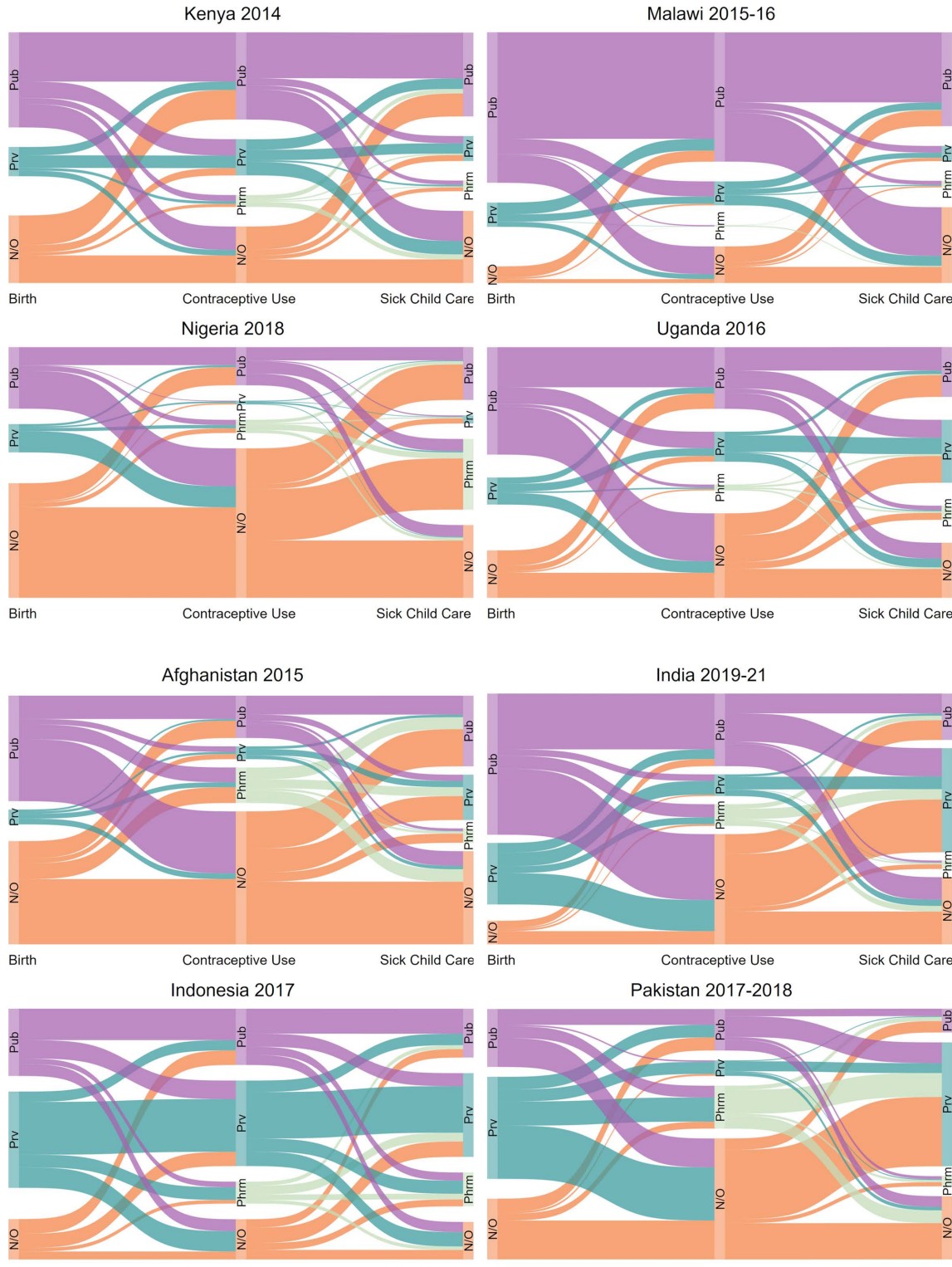

**Fig 1. A and B. Sector use from birth to contraception to sick child care among women with a need for all three services.** Note: Other/None for birth includes those who gave birth at home or other non-facility location; for contraceptive use, this includes informal sources, traditional method use, or no use; and sick child care includes informal sources or no care. Prv = Private; Pub = Public; Phrm = Pharmacy.

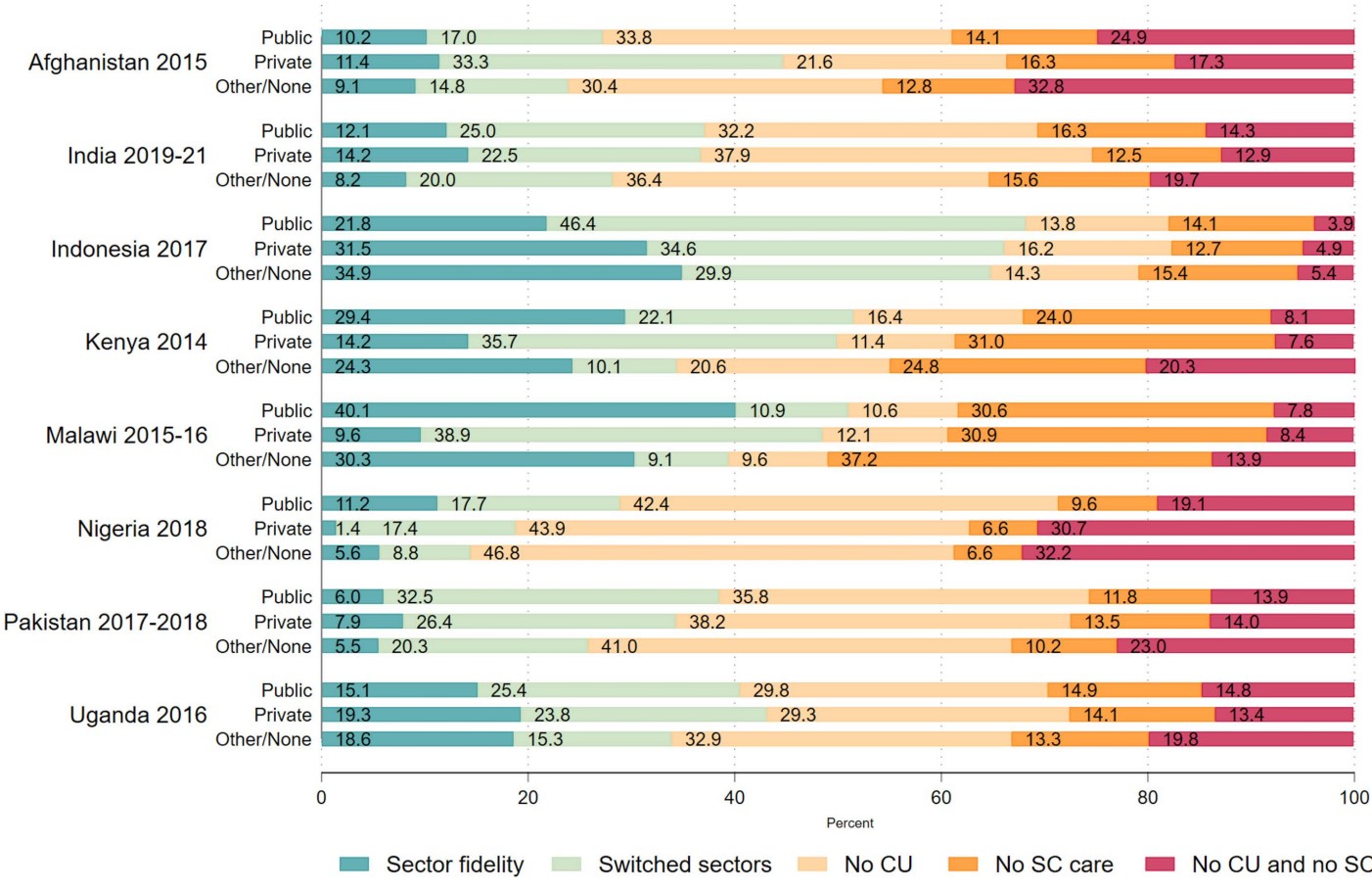

**Fig 2. Percentage of women with sector fidelity, sector switching, and missed opportunity for contraception and sick child care, by sectors and use of a facility for birth among women with a need for all three services.** Note: Sector fidelity refers to using the same sector for all three services, among those who gave birth in a facility, or the same sector for contraceptives and sick child care only for those who gave birth at home/other location. Other/None = includes those who gave birth at home or other non-facility location. CU = (Modern) contraceptive use; SC = Care for sick children.

33.6%). In Nigeria, missing opportunities for subsequent care was more common for women who gave birth in the private sector than in the public sector for both contraception (74.6% versus 61.5%) and sick child care (37.3% versus 28.7%). In all countries the percentage of those with a missed opportunity for contraception or sick child care was highest among those giving birth at home, though the difference was marginal in Indonesia.

Results of adjusted multivariable regression models are depicted in Fig 3 (Table A and Table B in S1 Tables) and represent the associations between sector of birth and subsequent missed opportunities for care for contraceptives or sick children in each country, independent of potential confounding variables. No clear pattern emerged across countries in the adjusted association of sector of birth—private sector vs. public sector—and subsequent missed opportunities for receiving contraceptives. In Afghanistan, delivery in a private sector facility was associated with lower odds of a subsequent missed opportunity for contraceptives (adjusted odds ratio [aOR]: 0.54, Confidence Interval (CI): 0.33-0.88) relative to delivery in the public sector. In India, Indonesia, and Nigeria there was a positive association between private sector delivery and missed opportunities for contraceptives (aOR: 1.23, CI: 1.12-1.36; aOR: 1.27, CI: 1.03-1.56; and aOR: 1.68, CI: 1.18-2.39, respectively). In the other four countries, there was no significant association between sector of delivery and missed opportunities for subsequent contraceptive use. Women who delivered at home or outside a facility had significantly higher adjusted odds

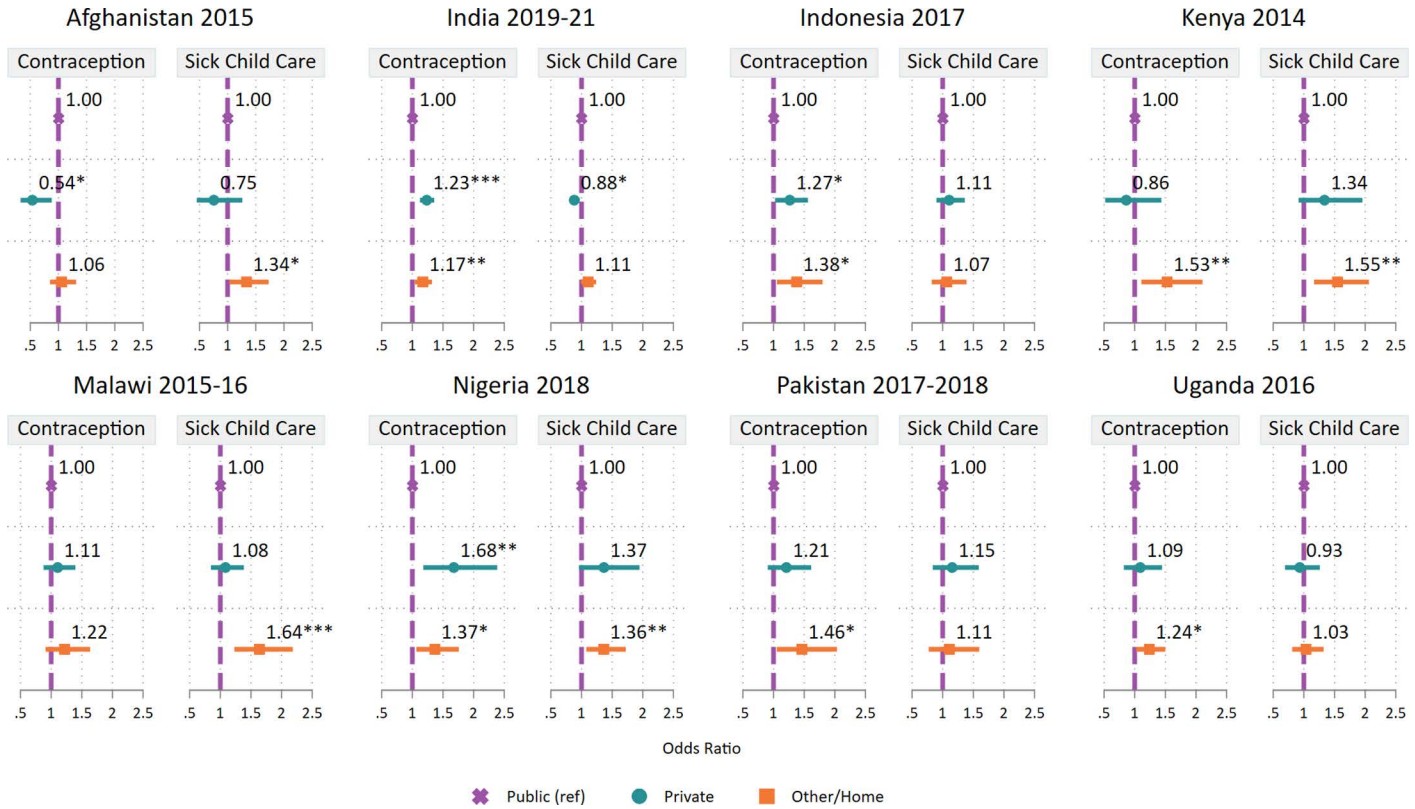

**Fig 3. Adjusted odds ratios and 95% confidence intervals of having no care for contraception or a sick child by place of birth among women with a need for all three services.** Note: Ref = reference.* p < 0.05 ** p < 0.01 *** p < 0.001.

of a subsequent missed opportunity for contraception relative to those who delivered in a public facility in six of the eight countries: India (aOR: 1.17, CI: 1.04-1.32), Indonesia (aOR: 1.38, CI: 1.06-1.80), Kenya (aOR: 1.53, CI:1.11-2.11), Nigeria (aOR: 1.37, CI: 1.07-1.76), Pakistan (aOR: 1.46, CI: 1.05-2.04), and Uganda (aOR: 1.24, CI: 1.02-1.50).

There was only one country where delivery in a private facility (relative to delivery in a public facility) was associated with lower odds of a subsequent missed opportunity for sick child care (India: aOR: 0.88; CI: 0.80-0.97). In four countries, birth outside of a health facility was positively associated with missed care for sick children: Afghanistan (aOR: 1.34, CI: 1.03-1.73), Kenya (aOR: 1.55, CI:1.17-2.06), Malawi (aOR: 1.64, CI: 1.23-2.18), and Nigeria (aOR: 1.36, CI: 1.08-1.72).

## Sector use patterns and subsequent missed opportunities for care

Results of adjusted multivariable regression models are depicted in Fig 4 (Table C in S1 Tables) and represent the associations between sector switching (between birth and contraceptive use) and missing sick child care, independent of potential confounding variables. In the only two countries with a significant association between switching sectors between birth and contraceptive use and missing sick child care, the direction of the association differed. In Pakistan, switching sectors was associated with a 72% increase in the odds of missing sick child care relative to those with sector fidelity (aOR: 1.72, CI: 1.05-2.84). In Nigeria, however, those who switched sectors had significantly lower odds of missing sick child care relative to those with sector fidelity (aOR: 0.57, CI: 0.33-0.99).

Missing opportunities for both facility birth and contraception was generally associated with also missing subsequent sick child care. When compared to women with sector fidelity, women who neither gave birth in a facility nor used modern

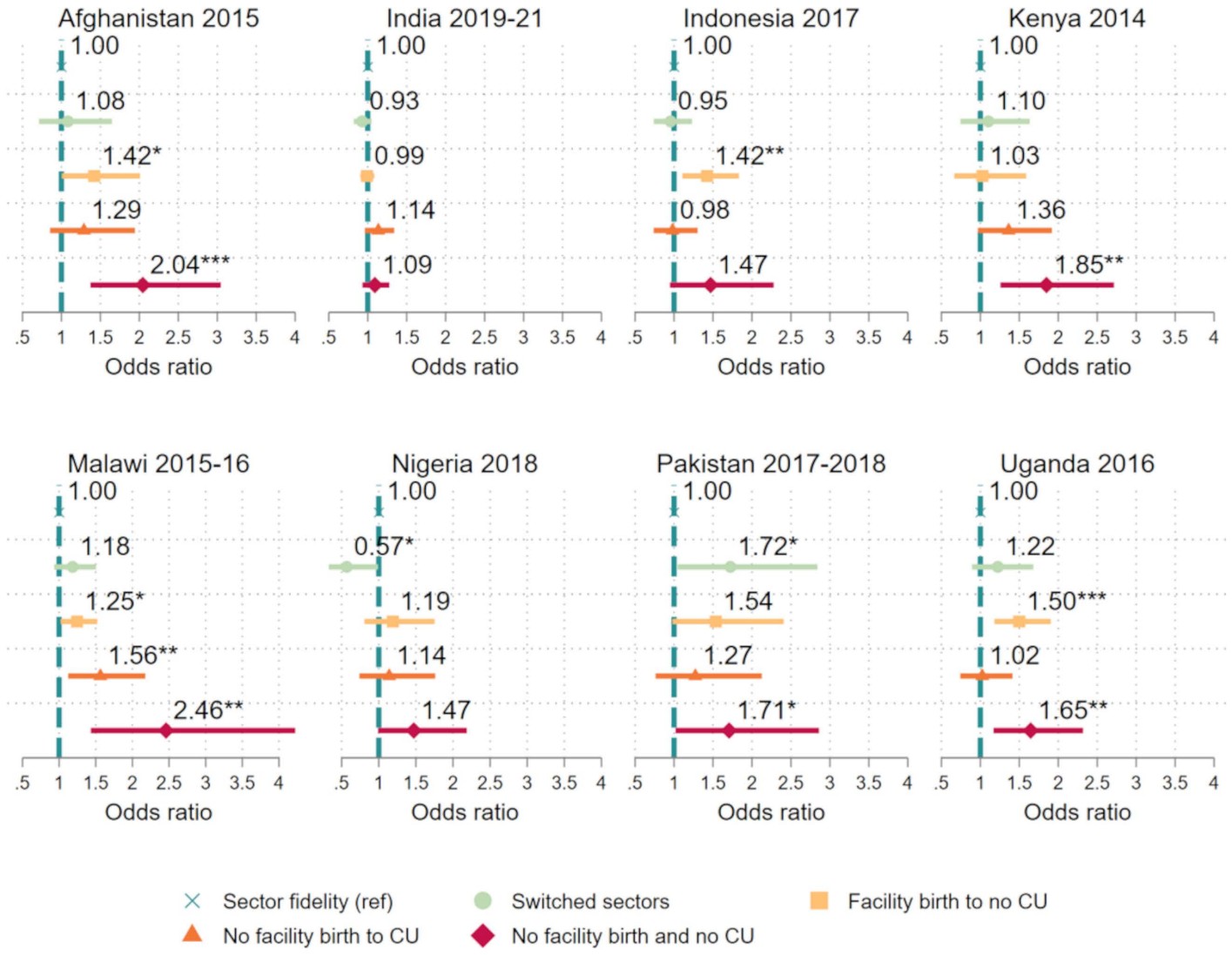

**Fig 4. Adjusted odds ratios and 95% confidence intervals for having missed sick child care by sector fidelity and nonuse of services for birth and contraception among women with a need for all three services.** Note: Ref = reference.* p < 0.05 ** p < 0.01 *** p < 0.001; CU = (Modern) contraceptive use.

contraception had significantly greater odds of missing sick child care in five of the eight countries (Afghanistan, Kenya, Malawi, Pakistan, and Uganda). In these countries, the adjusted odds ratios ranged from 1.65 (CI: 1.17-2.32) in Uganda to 2.46 (CI: 1.43-4.22) in Malawi.

Likewise, women with a facility birth who then missed an opportunity for contraception were also generally more likely to subsequently miss sick child care. There were significant positive associations between giving birth in a facility but not using modern contraception and then missing care for a sick child relative to sector fidelity in Afghanistan (aOR: 1.42; CI: 1.01-2.01), Indonesia (aOR: 1.42, CI: 1.11-1.83), Malawi (aOR: 1.25, CI: 1.02-1.52), and Uganda (aOR: 1.50, CI: 1.18-1.90).

On the other hand, women without a facility birth who then obtained contraception generally had similar odds of missing sick child care as compared to women with previous sector fidelity.

## Discussion

Our analysis found that switching between sectors was common across the eight countries, in line with Wong et al. [12], where switching sectors was common across maternal health care services. Our study expands upon this to include contraceptive use and examines subsequent missed opportunities, which were not strongly associated with dropping out of subsequent care.

This finding is important for two reasons. First, there is substantial evidence to show that the private sector is a significant source of care for women in LMICs, and particularly for contraception [11]. Considering the value of accessing services for birth, contraception, and sick child care, encouragingly, our analysis found that switching between the public and private sectors does not negatively correlate with future health care seeking. Secondly, our finding that sector switching was common underlines the importance of recognizing that most health systems are "mixed systems, where goods and services are provided both by the public and private sector, and health consumers are requesting these services from both sectors" [26]. Global strategies have called for greater recognition and leveraging the private sector within mixed health systems as essential to achieving national and Sustainable Development Goals [27]. Global agencies (such as donors) and governments seeking to leverage the private health sector in LMICs for improved health outcomes should strive to ensure that mechanisms are in place to foster private sector engagement and include these providers in health systems planning and management.

Our analysis also identified that sector fidelity with public facilities was more frequent in countries with large public sector dominance and, similarly, private fidelity was more common where the private sector has a large presence. Again, there may be many reasons behind an individual's decision to switch sectors during this continuum, such as the ability of the individual to exercise choice of sector or services in their country, or conversely, that the individual faces a lack of choice due to gaps in health care availability across the two sectors. As the private sector is typically accessed by those with means for choice (i.e., individuals from urban areas and/or in higher wealth quintiles) [8,10,13,14], the findings of this study add that public sector fidelity may be a function of lack of affordable or available private sector facilities. This area warrants further research, as existing evidence is unable to speak to the determinants behind these sector patterns.

Another striking finding of our analysis was how many women across the eight countries did not access the health system at any point in the continuum, or who dropped out of the continuum at later points. One approach to behavior change, the Integrated Gateway Model [28], postulates that one health behavior or 'gateway moment' may trigger or facilitate a subsequent behavior across the life cycle. In this way, interactions with health providers at one point in time can influence care in the future, especially at a gateway moment when an individual is more receptive to new information and sustained behavior change around health seeking, such as during pregnancy and childbirth [28]. These critical touch points also facilitate continuity of care through appropriate provider attention and referral for other health needs, facility coordination within networks, or physical integration or availability of multiple services within facilities [29,30]. For example, improved interpersonal communication and initiation of ANC in the first trimester was associated with greater health seeking for contraceptive and MNCH services during a woman's life course [31–33]. Greater use of maternal health care services, including both ANC and postnatal care but especially ANC, has demonstrated a positive relationship with modern contraceptive use after birth [34]. These behavioral gateway moments tend to bring women who might not otherwise seek or access care into the health system and therefore offer an opportunity to impact a woman and her family's current and future health [35].

Whether a woman switches sectors during this sequence of services is irrelevant if a woman never accesses formal health care at all. Both the public and private sectors can more effectively shape future care-seeking behavior at these critical gateway moments by ensuring consistently high-quality, person-centered care. Although the experience and quality of care vary across settings, with no sector consistently outperforming the other across countries, quality of care plays an important role in individuals' decision-making about choice and sector of facility [36]. Experiencing poor-quality care could understandably deter individuals from seeking future care [3,36]. However, significant improvements are needed in the quality of care across both sectors along the FP and MNCH care continuum [37–39].

 

Our analysis highlights that more important than the sector used or whether switching occurred, previous non-use of health facilities was a predictor of missed care. Ensuring access to high-quality health care, no matter the sector, must remain a priority for improving health outcomes. These findings underscore the need for health system stewards to closely consider both the public and private sectors in health governance. Greater cross-sectoral cooperation and continuity of care are paramount to improving health outcomes.

## Strengths and limitations

The strengths of this study lie in the multi-country analysis of health care use within unique healthcare systems while also identifying widespread trends. Using retrospective reports of health behavior, this study applied a life course perspective to the three services to understand care seeking dynamics to contextualize missed opportunities for care comparatively across sectors and across countries. To our knowledge, this is the first study to take such a life course approach to examine sector use and gaps in care across this sequence as they relate to sector use.

There are several important limitations. First, this study cannot specify whether sector fidelity reflects continuity of care by the same health providers or at the same health facilities, as women may still switch providers even when using facilities within the same sector. Second, our sample may include those without a true need for the health care services in question. The construct of need for contraception is dynamic, complex, and difficult to measure. The FP community of practice commonly defines need for FP based on a widely used but limited set of criteria [40]. As such, our sample may include women who are defined as having a need but have other reasons for contraceptive nonuse related to individual preferences rather than lack of access [40]. For sick child care, sick children may only have mild symptoms that resolved quickly and therefore did not also warrant a visit to a health care provider. The survey questions do not collect information that would allow for the use of a more precise denominator. As a result, the findings are likely biased towards the null.

Third, the sequence of services used in our analysis was selected based on available data as presented in the DHS, and our analysis does not approximate the entirety of care seeking for all services or for all who may seek care for each service independently. Using a different sequence of contraception, reproductive health and MNCH services, or different services altogether, may yield different results. Nulliparous, never-married, or adolescent and young women are also not captured; yet, this population, especially adolescents and young people aged 10–29, are an important population of study for the FP community of practice. Given that the current generation of adolescents aged 10–24 is the largest to date [41], and that approximately 90% of these young people live in LMICs [42], this population has an outsized effect on the future health and socio-economic well-being of entire countries. Additional research into patterns of care seeking among youth is needed for governments seeking to be effective stewards of mixed health systems. As these surveys were administered to women, and our sample included only women with a birth in the five years preceding the survey, other caretakers may have taken the sick child for care, which the mother may not have reported. Like much research based on self-reported survey data, recall and social desirability bias are likely to be present. Although questions about health service use, such as facility-based care and treatment for sick children, are relatively accurately reported [43,44], the ability of respondents to understand, classify, and recall the sector of service used may be limited [45]. Lastly, even though we did identify patterns between countries with large public sector dominance and larger private sector presence, the findings may not generalize to countries that were not studied in this analysis given the unique health systems in each country.

## Conclusion

To date, evidence of sector-specific patterns of use and future care seeking of specific services has been limited. Our study provided insight on the wide variation in patterns of sector use both across countries, and services within countries. The analysis also identified many opportunities for care being missed by both women and children, which were not typically associated with sector use and sector switching. Ideally, any interventions related to sector switching or sector fidelity should first be informed by the local context and an understanding of how service users navigate and interact with

the public and private sectors. With greater clarity on how individuals make health care decisions and how mixed health systems are, or are not, serving individuals' needs, system stewards can improve health outcomes by identifying and addressing gaps that may lead to missed opportunities for care. To inform these efforts, MPHD built an interactive data visualization tool using key findings from the analysis to allow the communities of practice to better engage with the data, accessible here: https://www.sectordynamics.org/

## Supporting information

**Table A in S1 Tables. Adjusted odds ratios and 95% confidence intervals for having a missed opportunity for use of modern contraceptives, comparing sectors and use of health services for birth and family planning among women with a need for birth, family planning, or sick child services.** Note: AOR: adjusted odds ratio; CI: confidence interval; Ref: reference; ANC: antenatal care. C-section: cesarean birth.
(DOCX)

**Table B in S1 Tables. Adjusted odds ratios and 95% confidence intervals for having missed oppportunity for care for sick children, comparing sectors and use of a facility for birth among women with a need for birth, family planning, or a sick child services.** Note: AOR: adjusted odds ratio; CI: confidence interval; Ref: reference; ANC: antenatal care. C-section: cesarean birth.
(DOCX)

**Table C in S1 Tables. Adjusted odds ratios and 95% confidence intervals for having missed care for sick children, comparing sectors and use of health services for birth and family planning among women with a need for birth, family planning, or sick child services.** Note: AOR: adjusted odds ratio; CI: confidence interval; Ref: reference; ANC: antenatal care; C-section: cesarean birth.
(DOCX)

## Acknowledgments

The authors would like to thank the team at the United States Agency for International Development (USAID) who supported MOMENTUM Private Healthcare Delivery to undertake this analysis and reviewed the findings. They would also like to thank Susannah Gibbs of PSI for their review of the report. We gratefully acknowledge support from the Eunice Kennedy Shriver National Center for Child Health and Human Development grant P2C-HD041041, Maryland Population Research Center.

## Author contributions

**Conceptualization:** Lindsay Mallick, Nicole Bellows, Rebecca Husband, Michelle Weinberger.

**Data curation:** Lindsay Mallick.

**Formal analysis:** Lindsay Mallick.

**Methodology:** Lindsay Mallick, Nicole Bellows.

**Project administration:** Rebecca Husband, Michelle Weinberger.

**Software:** Lindsay Mallick.

**Visualization:** Lindsay Mallick, Michelle Weinberger.

**Writing – original draft:** Lindsay Mallick, Nicole Bellows, Rebecca Husband, Michelle Weinberger.

**Writing – review & editing:** Lindsay Mallick, Nicole Bellows, Rebecca Husband, Michelle Weinberger.

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
