## [Decision Letter · Decision Letter 0]

28 Jan 2025

PGPH-D-24-01447

Dynamics of Care and Sector Use between Birth, Contraception and Sick Child Services

Dear Dr. Mallick,

Thank you for submitting your manuscript to PLOS Global Public Health. We apologize for the prolonged review period; identifying and securing peer reviewers was more difficult than usual during this period. Two reviewers have now provided detailed reviews. After careful consideration, we feel that the manuscript has merit but does not fully meet PLOS Global Public Health’s publication criteria as it currently stands. Therefore, we invite you to submit a revised version of the manuscript that addresses the points raised during the review process.

We look forward to receiving your revised manuscript.

Kind regards,

Hannah Tappis, DrPH, MPH

Academic Editor

Journal Requirements:

1.  Please provide an Author Summary. This should appear in your manuscript between the Abstract (if applicable) and the Introduction, and should be 150–200 words long. The aim should be to make your findings accessible to a wide audience that includes both scientists and non-scientists. Sample summaries can be found on our website under Submission Guidelines:

https://journals.plos.org/globalpublichealth/s/submission-guidelines#loc-parts-of-a-submission.

Additional Editor Comments (if provided):

Reviewers' comments:

Reviewer's Responses to Questions

**Comments to the Author**

1. Does this manuscript meet PLOS Global Public Health’s publication criteria ? Is the manuscript technically sound, and do the data support the conclusions? The manuscript must describe methodologically and ethically rigorous research with conclusions that are appropriately drawn based on the data presented.

Reviewer #1: Yes

Reviewer #2: Yes

2. Has the statistical analysis been performed appropriately and rigorously?

Reviewer #1: Yes

Reviewer #2: Yes

3. Have the authors made all data underlying the findings in their manuscript fully available (please refer to the Data Availability Statement at the start of the manuscript PDF file)?

Reviewer #1: Yes

Reviewer #2: Yes

4. Is the manuscript presented in an intelligible fashion and written in standard English?

Reviewer #1: Yes

Reviewer #2: Yes

5. Review Comments to the Author

Reviewer #1: Feedback on the Manuscript – Introduction and Method Section

Overall Impression:

The introduction provides a strong foundation for the study, effectively outlining the importance of understanding patterns of care seeking across the public and private sectors for family planning, reproductive health, and maternal, newborn, and child health services.

Specific Feedback:

• Strengthen the Thesis Statement: While the introduction provides a clear overview, consider adding a more focused thesis statement that directly states the research question or objective of the study. For example, "This study aims to examine patterns of sector use and sector fidelity for childbirth, modern contraceptive use, and sick child care among women in eight countries in Asia and sub-Saharan Africa."

• Discuss the Implications of Sector Switching: Elaborate on the potential implications of sector switching for continuity of care and health outcomes. For example, you could discuss the risks of missed opportunities for follow-up care or the potential for fragmented care.

• Highlight the Study's Contribution: Explain how the study will contribute to the existing literature on public-private health systems and women's health.

Results, Discussion and Conclusion Section

Overall Impression:

The results section provides a comprehensive overview of the study's findings, effectively presenting the patterns of sector use, sector switching, and missed opportunities for care. The use of tables and figures is helpful in visualizing the data.

Specific Feedback:

• Discuss the Implications of Sector Switching: While you've identified that sector switching is common, discuss the potential implications of this for continuity of care and health outcomes. For example, explore whether sector switching is associated with better or worse health outcomes compared to sector fidelity.

• Explore the Role of Quality of Care: Consider exploring the role of quality of care within the public and private sectors in influencing women's decisions to switch sectors or remain with a particular sector.

• Acknowledge Limitations of the Data: Acknowledge potential limitations of the DHS data, such as the reliance on self-reported information or the possibility of recall bias.

• Discuss Policy Implications: Explore the potential policy implications of the findings, such as recommendations for improving the coordination between public and private health systems or for investing in the quality of care in both sectors.

Reviewer #2: This is an interesting study looking at the use of private vs public sector use for childbirth, family planning and sick child care. I particularly find the "continuum of care" approach refreshing. I find it a bit challenging to lump countries in one big "LMIC" group for this topic as access to care and use of care, especially when it comes to women's health is heavily influenced by local culture. Your results for Afghanistan are interesting-- are you able to find references to explain them? If wonder if the sociopolitical context of the country affect women's ability to access to contraception within the public sector...

line 38. "in use of these three services" sounds awkward, peharps just "pattern in use" will suffice

line 47-49-- out of pocket cost and accessibility are not just factors for private sector, also affect use of public sector. are there studies that also look at reputation of public vs private sector as influencing factors?

line 57-62-- I disagree with these statements. Do you have references to support those statements? If not I would consider removing them or rewording them. In my experience, follow ups and referrals are made based on what the patient needs, they are not siloed within each sector. If a public facility does not offer a service, it will refer the patient to a private facility and vice versa-- whether the patient ends up accessing care depends on other factors such as easy access, affordability,... Same for sector fidelity-- does not mean much if patients access care at different facilities each time they need care as their records may not be available from one private facility to the next.

line 61-- why Asia and SSA in particular?

line 201-- please insert reference for Stata

line 214-- missing a word "more than ...resided"

215-217--unclear which percentages belong to which country

line 256-- reads "fig 1". did you mean fig 2?

figure 2 -- what does x axis represent?

lines 321-322-- "Considering the general desirability that women in need of delivery, contraception and sick child health services go on to access these services"-- awkward phrasing, please re-write

line 338-- remove the "." before "the findings"

line 344- please insert reference for the Integrated Gateway Model

Strengths and limitations-- should mention retrospective nature of data introduces chances of recall bias. Also, sick child care might not be the sole responsibility of the mother-- father or other family member could have taken the child to access care. In some communities, there are community healthcare workers who may be able to take care of minor illnesses-- was that captured in the survey?

6. PLOS authors have the option to publish the peer review history of their article (what does this mean? ). If published, this will include your full peer review and any attached files.

**Do you want your identity to be public for this peer review?** For information about this choice, including consent withdrawal, please see our Privacy Policy .

Reviewer #1: **Yes: ** Abdulmalik Alilu Abubakar

Reviewer #2: No

---

## [Editor Report · Decision Letter 1]

3 Mar 2025

Dynamics of Care and Sector Use between Birth, Contraception and Sick Child Services

PGPH-D-24-01447R1

Dear Mallick,

We are pleased to inform you that your manuscript 'Dynamics of Care and Sector Use between Birth, Contraception and Sick Child Services' has been provisionally accepted for publication in PLOS Global Public Health.

Best regards,

Hannah Tappis, DrPH, MPH

Academic Editor